# Physcion-Matured Dendritic Cells Induce the Differentiation of Th1 Cells

**DOI:** 10.3390/ijms21051753

**Published:** 2020-03-04

**Authors:** Yun-Ho Hwang, Su-Jin Kim, Sung-Tae Yee

**Affiliations:** College of Pharmacy, Sunchon National University, 255 Jungangno, Suncheon 540-950, Korea; hyh7733@naver.com (Y.-H.H.); ksz1353@naver.com (S.-J.K.)

**Keywords:** Physcion, DCs, CD4^+^ T cells, helper T cells, asthma

## Abstract

In addition to their use as colorants, anthraquinone derivatives have numerous medical applications, for example, as antibacterial and antiinflammatory agents. We confirmed that physcion (an anthraquinone derivative) induces TNF-alpha production by macrophages and increased the expressions of surface molecules (CD40, CD80, and CD86) and major histocompatibility complex (MHC) II. Based on these results, we hypothesized that physcion might induce the maturation of dendritic cells (DCs) to antigen-presenting cells (APCs), and decided to conduct in vitro experiments using bone-marrow-derived DCs (BMDCs). Physcion was not toxic to DCs and increased the expression of surface molecules (e.g., CD40, CD80, CD86, and MHC II) and the production of cytokines (e.g., IL-12p70, IL-1beta, IL-6, and TNF-alpha), but not of IL-10. To confirm that DCs matured by physcion induce T-cell-immune responses, naive CD4^+^ T cells were treated with physcion-treated DCs or their supernatants. Physcion induced the maturation of DCs, which promoted the polarization of Th1 cells. Our results show physcion-induced DC maturation via TLR4, and that mature DCs promote the differentiation of Th1 cells without affecting the differentiation of Th2 cells. These findings show that physcion has potential use as a treatment for inflammatory diseases associated with Th1/Th2 cell imbalance.

## 1. Introduction

Asthma affects more than 5% of the world′s population, and its prevalence continues to increase, especially among children. Despite recent advances in discovery and treatment, asthma remains a cause of serious morbidity and economic burden [1]. Asthma is characterized by airway inflammation and increased mucus secretion and bronchial responses to various stimuli and causes intermittent symptoms such as wheezing, coughing, and dyspnea [2]. Inflammatory response in asthma is associated with eosinophils, neutrophils, basophils, mast cells, and Th2 lymphocytes (CD4^+^IL-4^+^ T cells), which are found in the lungs of asthmatic patients [3]. It is known that, in the allergic asthma airway, inflammation occurs when the Th2 cell-mediated immune response is greater than the Th1 cell-mediated immune response. Similarly, Th1 and Th2 cell-mediated immune responses are of similar strengths, which suggests that excessive Th1 cell-mediated immune responses can cause asthma-like symptoms [4]. Accordingly, it has been suggested that regulating Th1/Th2 cell balance might offer a means of treating asthma without harmful side effects [5].

Dendritic cells (DCs) are the antigen-presenting cells (APCs) responsible for the initiation and regulation of immune response and are efficient stimulators of B and T lymphocytes. B cells, which are precursors of antibody-secreting cells, can recognize intrinsic antigens directly using B cell receptors (BCRs), but T lymphocytes must recognize antigens presented by APCs. The T cell receptor (TCR) of T helper (Th) cells is activated by recognizing antigen fragments bound to the major histocompatibility complex (MHC) on APC surfaces [6], whereas TCR activation occurs in certain cytokine environments. Naive CD4^+^ T cells can differentiate into one of several lines, including Th1, Th2, Th17, or Treg [7]. Regulation of the expressions of specific cytokines secreted by DCs suggests the therapeutic possibility of inducing specific helper T cell differentiation to suppress the Th1/Th2 cell imbalance in favor of Th2 cells in allergic asthma.

Rhubarb and aloe contain anthraquinone and are used as folk remedies. Biologically active anthraquinone derivatives are found in bacteria, fungi, and insects, and are used in a variety of products such as textile dyes, paints, foods, cosmetics, and pharmaceuticals [8]. Furthermore, anthraquinone derivatives have various biological effects such as anti-cancer [9], anti-inflammatory [10], antibacterial [11], and antiviral effects [12]. Physcion is an anthraquinone derivative and has been shown to induce apoptosis of acute lymphoblastic leukemia [13], human nasopharyngeal carcinoma [14], human colorectal cancer (SW620 cells) [15], and hepatocellular carcinoma cells [16]. Physcion 8-O-β-glucopyranoside reduces the release of pro-inflammatory cytokines by downregulating the expression of Toll-like receptor (TLR) 2 and TLR4, and these down-regulations have been reported to have an anti-septic effect [17]. Although physcion has been reported to have various biological activities, its effects on the maturation of DCs have not been previously studied.

In this study, we examined the effect of physcion on the maturation of DCs and the effect of these mature DCs on T cell differentiation with a view toward investigating the possible use of physcion for modifying T cells in asthma.

## 2. Results

### 2.1. Effect of Physcion on Cytotoxicity

To evaluate the cytotoxic effects of physcion on BMDCs, we counted dead cell numbers (Figure 1A). We found that Annexin V^+^/PI^+^ populations in the physcion 1 µM (23.2% ± 0.14%), 10 µM (24% ± 2.55%), and 100 μM (25% ± 3.75%) groups did not differ from that in the control group (27.2% ± 1.98%) (Figure 1B). The relatively high cell deaths observed were attributed to the time required for antibody staining.

### 2.2. Effect of Physcion on the Maturation of BMDCs

To determine if physcion induces maturation, we assessed the levels of the BMDC maturation markers CD40, B7 (CD80 and CD86), and MHC II (major histocompatibility complex II) (Figure 2A) using LPS as a positive control. CD40 levels were significantly and concentration-dependently increased by physcion versus non-treated controls (Figure 2B), and expressions of CD80 (Figure 2C), CD86 (Figure 2D), and MHC II (Figure 2E) were also significantly increased by physcion at 10 and 100 μM. In addition, CCR7 expression, which is involved in DCs movement, was significantly increased by LPS or physcion (Figure 3A). Moreover, endocytic activity (an active marker of mature DCs) was assessed by exposing cells to dextranfluorescein isothiocyanate (FITC). Double-positive cell (CD11c^+^ and dextran-FITC-positive) numbers were decreased by physcion or LPS, indicating that functional maturity was enhanced (Figure 3B). These results show that physcion increased the expressions of the co-stimulatory molecules required to produce activated CD4^+^ T cells from DCs and enhanced functional maturity.

### 2.3. Effects of Physcion on the Production of Cytokines by BMDCs

B and T lymphocytes act as mediators in the immune system, but their functions are mediated by DCs. DCs that capture an antigen express co-stimulatory molecules and produce various cytokines. Cytokines secreted by DCs markedly influence T cell immunity. In particular, IL-12 promotes the differentiation of Th1 cells, which results in the production of IFN-γ. [6]. To confirm that physcion induces DCs to produce cytokines, we assessed cytokine levels in the culture media of DCs treated with LPS or physcion by ELISA. Physcion of LPS treated BMDCs excreted significantly higher levels of IL-12p70, IL-1β, IL-6, and TNF-α to media than non-treated controls (Figure 4A–D). However, IL-10 levels remained at the control group level (Figure 4E). These results suggest that DCs matured by physcion induce T-cell-mediated immune responses.

### 2.4. Effects of Physcion-Stimulated DCs on the Differentiation of Effector T Cells

To confirm the effect of physcion-maturated BMDCs on the differentiation of Th1 or Th2 cells, we co-cultured LPS or physcion-activated BMDCs and naive CD4^+^ T cells and then analyzed populations of Th1 and Th2 cells by FACS. The supernatants of BMDCs (Figure 5A) or physcion-matured BMDCs (Figure 5B) induced T cell proliferation. To determine whether the T cell increase was due to effector T cells, we examined doubly positive CD4 and CD44 cell populations. We found that LPS or physcion significantly induced the differentiation of naïve CD4^+^ T cells to effect cells (Figure 6). These results indicate that physcion increases the differentiation of effector cells.

### 2.5. Effects of Physcion-Stimulated DCs on the Differentiation of Naive CD4^+^ T Cells

Physcion-maturated BMDCs and physcion-maturated BMDC supernatants did not promote the differentiation of naive CD4^+^ T cells to Th2 cells (Figure 7A), but both significantly increased the differentiation of naive CD4^+^ T cells to Th1 cells (Figure 7B,C).

### 2.6. Effect of TLR4 on Physcion-Induced DC Maturation

To examine the effect of TLR4 on physcion-induced DC maturation, we blocked TLR4 using an anti-TLR4 antibody. Treatment with LPS or physcion increased the expressions of CD80, CD86, and MHC2 in DCs as compared with non-treated controls, but knockdown of TLR4 significantly inhibited these treatment-induced increases in the expressions of CD80, 86, MHC2 by LPS or physcion (Figure 8A–C). In addition, increases in IL-12p70 induced by physcion or LPS were significantly decreased by TLR4 knockdown (Figure 8D). Moreover, the treatment of mature TLR4 knocked-down DCs with physcion markedly inhibited physcion-induced differentiation to Th1 cells (Figure 8E).

## 3. Discussion

CD4^+^ T cells play a crucial role in the control asthma-related inflammation and are the predominant lymphocytes that infiltrate airways [18]. CD4^+^ T cells are classified as Th1 or Th2 cells; Th1 cells produce IFN-γ, whereas Th2 cells produce IL-4, IL-5, and IL-13 [19]. Th2 cells predominate in the airways of patients with allergic asthma and are characterized by the cytokines they secrete [20]. The activation and differentiation of T cells require cytokines, co-stimulatory molecules, and major histocompatibility complex (MHC) complexes expressed by DCs, and thus the density and quality of DCs probably determines the magnitude and type of T cell response [21]. In this study, we tested the possibility of treating asthma by controlling the differentiation of CD4^+^ T cells via DC activation as an alternative to conventional asthma therapy.

Before performing in vitro experiments with DCs, we used RAW 264.7 cells as antigen presenting cells (APCs), to verify the activity of physcion, and measured the expressions of surface molecules. Physcion (1-100 uM) had no cytotoxic effect on macrophages (Appendix A), but at 100 uM increased the expressions of CD40, CD80, CD86, and MHCII (Appendix A), and thus was used at this concentration for in vitro experiments. Furthermore, TNF-α (a representative macrophage cytokine) was significantly increased by physcion treatment (Appendix A). These results suggest that physcion induces DC maturation. CD40 is a marker of mature DCs and CD40 signaling, leading to more effective antigen presentation due to the upregulation of MHC class Ⅱ and co-stimulatory molecules (CD80, CD86) [22]. Increased expression of CD40 molecules indicates an increased frequency of binding to CD40 ligand, which contributes to IL-12 production and the activation of T cells [23]. CD80 (B7-1), CD86 (B7-2), and MHC II (all markers of DC maturation) are increased by LPS [24]. In order for T cells to be optimally activated, recognition of Antigen / MHC complex by TCR (T cell receptor) requires ancillary stimulation signals that are provided by the associated co-stimulatory molecules CD80 (B7-1) and CD86 (B7-2) [25]. In the present study, we confirmed that CD40, CD80, CD86, and MHC class Ⅱ levels on LPS or physcion-treated DCs were higher than in non-treated controls. The up-regulation of CD40 in DCs by physcion seems to be related to physcion-induced increases in CD80, CD86, and MHC class Ⅱ expression, and their inductions of T cell differentiation and activity. Thus, these results indicate that physcion induces the maturation of DCs and the differentiation of naïve CD4^+^ T cells into helper T cells. Other cytokines influence the differentiation of CD4^+^ T cells to Th1 or Th2 cells. In particular, IL-12, which is secreted by antigen-presenting cells, including DCs, is involved in the induction of the differentiation of naive CD4^+^ T cells to Th1 cells [26]. On the other hand, IL-10 is involved in the differentiation of Th2 cells [27]. The present study shows that physcion increased the production of IL-12 by DCs but did not increase IL-10 production, which suggests that DCs matured by physcion stimulate the differentiation of T cells and that cytokines secreted by these cells can promote or inhibit T cell differentiation. We hypothesized that increased IL-12 secretion induced by physcion would support the differentiation of naive CD4^+^ T cells to Th1 cells.

High levels of IFN-gamma secreted by Th1 cells promote the differentiation of Th1 cells, while low concentrations of IFN-gamma promote Th2 responses [28]. When we co-cultured physcion-matured DCs cultured with naive CD4^+^ T cells, proliferating CD4^+^ T cells were found to be Th1 cells. In addition, co-cultivation of naive CD4^+^ T cells with physcion-matured DC supernatants resulted in a decrease in Th2 cells and an increase in Th1 cells. These results indicate that physcion promotes the expression of co-stimulatory molecules and the production of IL-12 by DCs, and that these promote the differentiation of Th1 cells.

Members of the Toll-like receptor (TLR) family recognize several microbial products. These receptors are characterized by extracellular leucine-rich domains, which are involved in the recognition of extracellular agents [29]. TLR2 and TLR4 stimulate cytokine production by DCs in an adapter protein- and myeloid differentiation factor 88 (MyD88)-dependent manner. Although TLR2 and TLR4 agonists differ, both agonists induce MyD88-dependent activity [30]. In the present study, we used anti-TLR2 and anti-TLR4 antibodies to confirm that physcion induces DC maturation through TLR4 activity. We observed that physcion failed to induce the maturation of anti-TLR4 antibody treated DCs, and co-culture with these cells did not induce naive CD4^+^ T cells to differentiate into Th1 cells. On the other hand, physcion induced the maturation of anti-TLR2 antibody-treated DCs. These results indicate that physcion-induced DC maturation is associated with TLR4.

Inhaled corticosteroids are used in many countries to treat chronic asthma [31]. Based on the GINA (Global INitiative for Asthma) guidelines, asthma treatment proceeds in five steps. Steps 1 to 4 primarily involve the use of low-dose inhaled glucocorticosteroids, whereas, in Step 5, oral glucocorticosteroids are administered [32]. Suissa S et al. reported the regular use of low-dose inhaled corticosteroids can meaningfully reduce asthma-associated hospitalization and mortality [33]. However, although they offer the most effective means of treating asthma, accumulated evidence shows that long-term corticosteroid oral administration is associated with increased risks of psychiatric disorders, infections, diabetes, and osteoporosis [34], and central nervous system complications, ophthalmological complications, cardiovascular complications, respiratory complications, gastrointestinal complications, renal complications, musculoskeletal complications, and hematological and immunological complications [35]. In order to eliminate these side effects, it is necessary to develop new asthma treatment drugs. Physcion tests the effects on the acute and chronic toxicity of various organs in the body and as a result, if it is not toxic, we think physcion is a good candidate for asthma treatment through Th1 / Th2 balance control.

## 4. Materials and Methods 

### 4.1. Reagents

Recombinant mouse granulocyte-macrophage colony-stimulating factor (GM-CSF), interleukin-4 (rmIL-4), and recombinant human interleukin-2 (rhIL-2) were purchased from R&D Systems for DC differentiation. Physcion, lipopolysaccharides (LPS), propidium iodine (PI), mitomycin C (MMC), phorbol 12-myristate 13-acetate (PMA), and inomycin were purchased from Sigma-Aldrich. The following FITC-, PE-, and PE-Cy5-conjugated monoclonal antibodies (Abs) were purchased from BD Biosciences; FITC-annexin Ⅴ, CD3 (145-2C11), CD28 (37.51), CD16/32 (2.4G2), CD11c (HL3), CD25 (PC61.5), CD40 (HM40-3), CD80 (16-10A1), CD86 (GL1), Ⅰ-A[b] (AF6-120.1), IL-4 (11B11), and IFN-γ (XMG1.2). Purified and biotinylated antibodies for murine IL-1β, IL-6, IL10, IL-12p70, and TNF-α were purchased from BD Biosciences [36].

### 4.2. RAW264.7 Cells Culture

RAW264.7 cells (a murine macrophage cell line) were purchased from the American Type Culture Collection (ATCC, Manassas, VA, USA), and cultured in RPMI1640 medium supplemented with 10% heat-inactivated FBS, 1% streptomycin/penicillin at 37  °C in a humidified 5% CO2 atmosphere. Introduce cells (5 × 105 cells/mL) were seeded in 24-well plates and stimulated with LPS (1 μg/mL) or physcion (1-100 μM) at 37 °C for 24 h in the same medium. Cell viability was measured using a CCK-8 kit and cytokine and surface molecule levels were measured the same way as DCs.

### 4.3. Differentiation of Bone Marrow Derived-DCs (BMDCs)

Seven-week-old female C57BL/6 and BALB/c mice (18–20 g) were purchased from Orientbio (Orientbio Inc., Iksan, Korea). Mice were housed in a controlled environment (22 ± 2 °C; 50 ± 5% RH) in polycarbonate cages and fed a standard animal diet with free access to commercial rodent chow (DAE-HAN Biolink, Daejeon, Korea) and water. Bone marrow cells (1 × 106 cells/mL/well) isolated from tibias and femurs of C57BL/6 mice were cultured in RPMI medium containing 10% FBS, 2-mercaptoethanol (50 μM/mL), IL-4 (1000 U/mL), and GM-CSF (1000 U/mL) in 6-well plates. On culture days 2 and 4, non-adherent cells were removed and fresh medium (containing cytokines) was added. On day 6, immature DCs were harvested and used for DC activity and T cell differentiation experiments.

### 4.4. Cytotoxicity Assay

BMDCs (1 × 106 cells/2 mL/well) were incubated for 18 h with physcion (1–100 μM) in 24-well plates. Cells were then treated with FITC-Annexin V in the dark for 15 min at room temperature (RT), and then with PI (Sigma, Cat# P-4864-10ML) in the dark for 15 min at RT.

### 4.5. Flow Cytometry

BMDCs treated with LPS (l μg/mL) or physcion (1, 10, 100 μM) for 18 h. Matured BMDCs were then harvested and washed with FACS buffer. In order to block FcγII and FcγIII on BMDCs, BMDCs were incubated with anti-CD16/32 for 30 min on ice. Cells (1 × 106 cells/mL) were then stained with fluorescence-labeled antibodies, that is, anti-mouse CD11c-FITC, CD40-PE, CD80-PE, CD86-PE, MHCⅡ-PE, and CCR7-PE (all 1 μg) for 30 min, washed with FACS buffer, and analyzed using a FACS CantoII unit. Forward versus side scatter (FSC vs SSC) gating parameters were used to identify cells of interest based on size and granularity (complexity).

### 4.6. Antigen Uptake Ability of BMDCs by Physcion

BMDCs (2 × 105 cell) were equilibrated at 37 or 4 °C for 2 h and then pulsed with fluorescein conjugated dextran at a concentration of 10 μg/mL. The reaction was stopped by adding cold stain buffer. Cells were then washed twice, stained with PE-conjugated anti-CD11c antibodies, and analyzed with the FACS CantoII unit.

### 4.7. Cytokine Assay

Cytokine levels in the supernatants of BMDCs treated with LPS or physcion for 18 h were measured by ELISA. The lower detection limits of these assays were 1.11 pg/mL for IL-6, IL-10, TNF-α, and IL-1β and were 3.9 pg/mL for IL-12p70. The first antibody (2 μg/mL) was incubated for 12 h at 4 °C. ELISA wells were blocked using an ELISA blocking buffer. Prepared samples and standards were dispensed into wells and incubated at RT for 4 h. The biotin-conjugated secondary antibody (1 μg/mL) is treated and cultured at RT for 1 h. After washing, avidin horseradish peroxidase was treated and incubated at RT for 1 h. The substrate was treated and the absorbance at 405 nm was measured via a microplate reader. IL-1β (BD Biosciences, San Diego, CA) and IL-12p70 (BD Biosciences, San Diego, CA) levels were measured according to protocols provided by the manufacturers.

### 4.8. In Vitro Priming of Naïve CD4+ T Cells

Naïve CD4+ T cells were isolated from splenocyte suspensions of BALB/c mice using the CD4+ T-cell isolation kit II (MACS; Miltenyi Biotec) and primed by treating them (1 × 105 cells/2mL/well) with 1 μg/mL of anti-CD3 and MMC-treated mature BMDCs (1 × 106 cells/2mL/well) in 24-well plates. Primed cells were cultured in RPMI1640 supplemented with 10% FBS and 1% penicillin and streptomycin and maintained in 5% CO2 humidified atmosphere at 37 °C for 4 days. The CD4+ T cells so obtained were expanded in four wells with fresh medium containing 2.5 ng/mL of rhIL-2 and cultured for another 2 days but, during the last 3 h, culture were re-stimulated with PMA (50 μg/mL) and ionomycin (1μM) for 5h in the presence of brefeldin A (5 mg/mL). Cells were then assayed for intracellular cytokines by FACS. Briefly, cells were stained with PE-Cy5-anti-CD4, fixed and permeabilized with 4% Fixation/Perm buffer Ⅲ, and stained with FITC-anti-IFN-γ and PE-anti-IL-4 [36]. The CD4+ T cell proliferation was measured by CCK-8 agent.

### 4.9. Statistical Analysis

Significances of intergroup differences were determined by one-way analysis of variance (ANOVA) with Duncan’s multiple range test using SPSS version 22 (Chicago, IL, USA). Results are presented as means ± SDs, and statistical significance was accepted for p values < 0.05.

## 5. Conclusions

In general, the polarization of Th1 or Th2 cells results in a variety of immunological diseases. Notably, in allergic asthma, the population of Th2 cells is greater than that of Th1 cells. IFN-gamma secreted by Th1 cells inhibits the differentiation of Th2 cells, whereas IL-4 secreted by Th2 cells inhibits the differentiation of Th1 cells. Furthermore, IL-12p70 secreted by DCs promotes the differentiation of Th1 cells, and thus induces DCs to produce IL-12p70, which presents a strategy for treating asthma based on promoting the differentiation of Th1 cells. In the current study, physcion increased the expressions of co-stimulatory molecules involved in the activation of T cells, and the IL-12 secreted by physcion-matured DCs contributed to the differentiation of Th1 cells. From a therapeutic point of view, it appears that the physcion-induced up-regulation of Th1 immune response offers a potential alternative means of treating asthma.

## Figures and Tables

**Figure 1 ijms-21-01753-f001:**
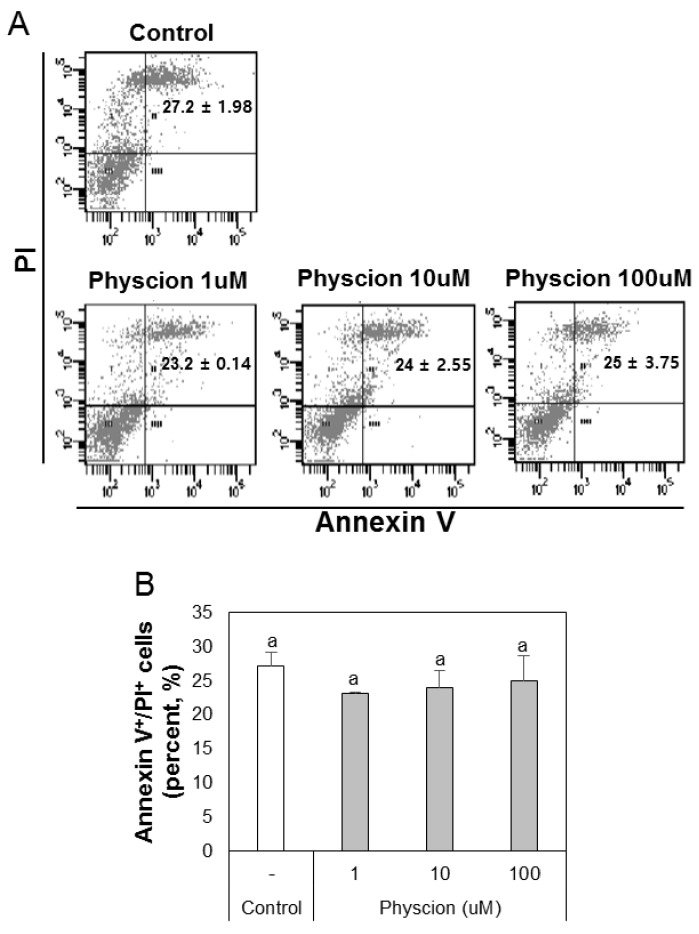
Physcion was not cytotoxic to bone-marrow-derived dendritic cells (BMDCs). (**A**) Cell death analysis by flow cytometry. Late-stage apoptotic cells are presented (Annexin V-FITC positive and PI positive). (**B**) Calculated apoptotic rates (%) of late stage apoptotic cells. Experiments were repeated twice. a: different letters indicate significant differences (*p* < 0.05) as determined by one-way ANOVA with Duncan’s multiple-range test.

**Figure 2 ijms-21-01753-f002:**
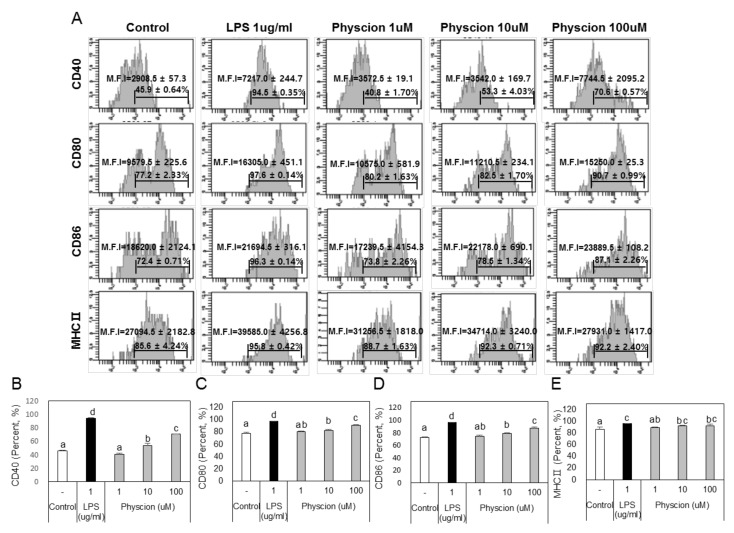
Physcion increased the expressions of surface molecules and MHCIIon BMDCs. (**A**) Histograms show percentages and mean fluorescence intensities (MFIs) of CD11c + CD40 +, CD11c + CD80+, CD11c + CD86+, and CD11c + MHCII+ cells. Bar graphs show calculated percentages of (**B**) CD11c + CD40+, (**C**) CD11c + CD80, (**D**) CD11c + CD86, and (**E**) CD11c + MHCIIcells. Experiments were repeated twice. a, b, c, and d: different letters indicate significant differences (*p* < 0.05) as determined by one-way ANOVA with Duncan’s multiple-range test.

**Figure 3 ijms-21-01753-f003:**
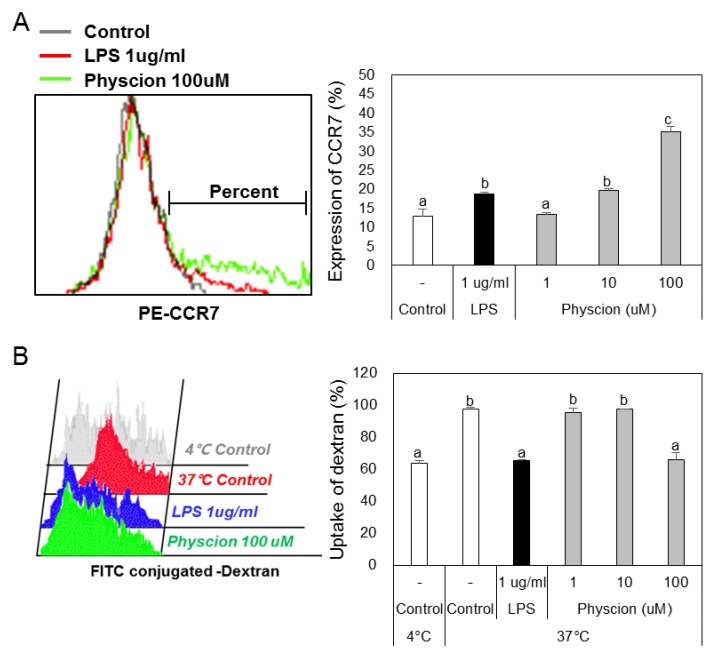
Physcion induced the functional maturation of BMDCs. (**A**) Percentages of CD11c + CCR7+ BMDCs and (**B**) percentages of dextran (FITC)-positive CD11c (PE)-positive cells are indicated. Experiments were repeated twice. a, b, c, and d: different letters indicate significant differences (*p* < 0.05) as determined by one-way ANOVA with Duncan’s multiple-range test.

**Figure 4 ijms-21-01753-f004:**
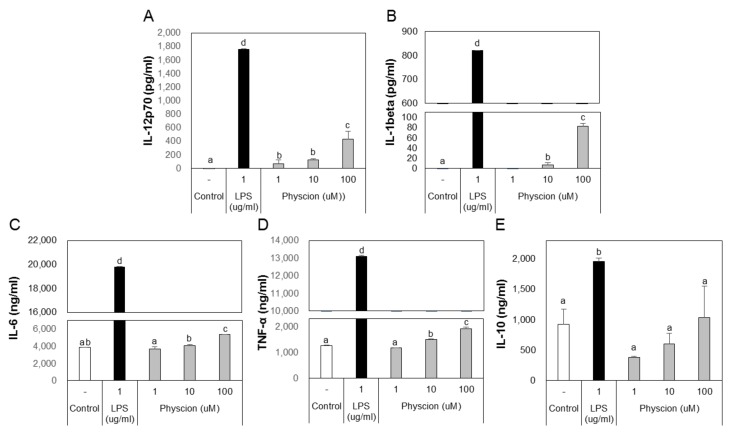
Physcion increased cytokine secretions by BMDCs. ELISA analysis of the cytokines, (**A**) IL-12p70, (**B**) IL-1 beta, (**C**) IL-6, (**D**) TNF-α, and (**E**) IL-10, secreted by LPS or physcion-treated BMDCs. Experiments were repeated twice. a, b, c, and d: different letters indicate significant differences (*p* < 0.05) as determined by one-way ANOVA with Duncan’s multiple-range test.

**Figure 5 ijms-21-01753-f005:**
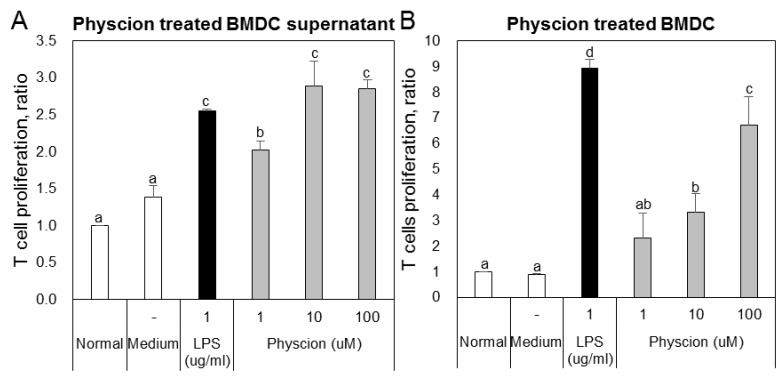
Physcion-treated BMDCs or supernatants induced T cell proliferation. Proliferations of CD4+ T cells co-cultured with (**A**) physcion-treated BMDCs or (**B**) their supernatants. Experiments were repeated twice. a, b, c, and d: different letters indicate significant differences (*p* < 0.05) as determined by one-way ANOVA with Duncan’s multiple-range test.

**Figure 6 ijms-21-01753-f006:**
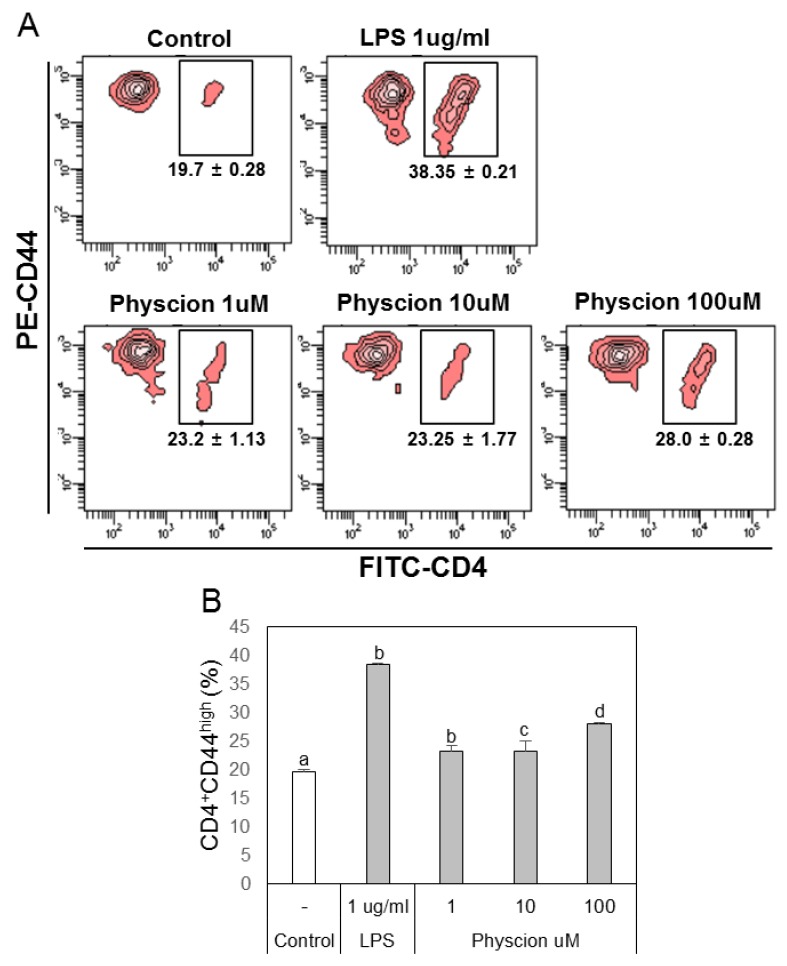
Physcion-treated BMDCs induced the differentiation of effector T cells (CD4 + CD44high cells). (**A**) Representative dot plots of CD4 + CD44high cells obtained from CD4+ T cells co-cultured with physcion-treated BMDCs. (**B**) Bar graphs indicate calculated percentages of CD4 + CD44high cells. Experiments were repeated twice. a, b, c, and d: different letters indicate significant differences (*p* < 0.05) as determined by one-way ANOVA with Duncan’s multiple-range.

**Figure 7 ijms-21-01753-f007:**
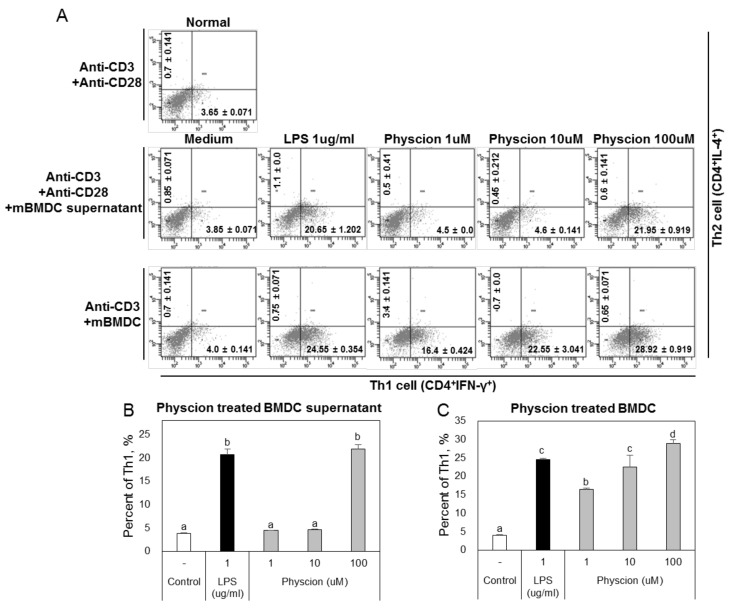
Physcion-treated BMDCs and their supernatants induced the differentiation of Th1 (CD4+ IFN-γ+) cells. (**A**) Representative dot plots of CD4+ IFN-γ+ and CD4+ IL-4+ cells of naïve CD4+ T cells co-cultured with physcion-treated BMDCs. Percentages of Th1 cells that differentiated from naive CD4+ T cells cultured with (**B**) physcion-treated BMDCs or (**C**) their supernatants. Experiments were repeated twice. a, b, c, and d: Different letters indicate significant differences (*p* < 0.05) as determined by one-way ANOVA with Duncan’s multiple-range test.

**Figure 8 ijms-21-01753-f008:**
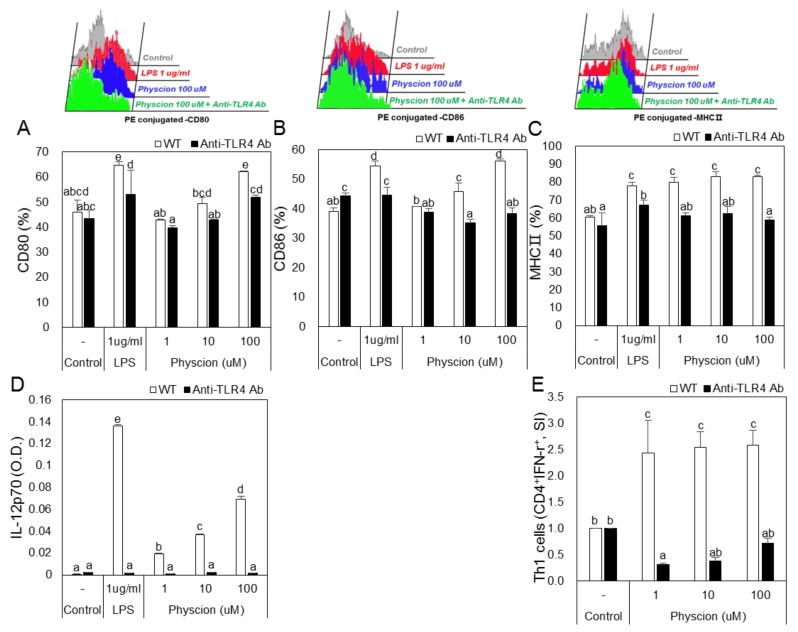
Blocking of TLR4 in BMDCs decreased surface molecule expression, cytokine production, and the Th1 cell population. Anti-TLR4 Ab treated or untreated BMDCs were incubated with physcion. Bar graphs shown calculated percentages of (**A**) CD11c + CD80+, (**B**) CD11c + CD86 +, and (**C**) CD11c + MHCII+ cells. (**D**) IL-12p70 levels were measured by ELISA and (**E**) Th1 cell populations were determined by FACS and are presented as CD4 + IFN- γ + cells percentages. Experiments were repeated twice. a, b, c, d, and e: Different letters indicate significant differences (*p* < 0.05) as determined by one-way ANOVA with Duncan’s multiple-range test.

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
