# Peer review of "Physcion-Matured Dendritic Cells Induce the Differentiation of Th1 Cells"

_ijms, 2020, doi:10.3390/ijms21051753_

Round 1

Reviewer 1 Report

Abstract

Please provide a one or two-sentence rationale and hypothesis for testing the effects of physcion on dendritic cells.  Currently, the rationale for testing is not obvious.

Lines 22-24; Please simplify the following sentence, as it is difficult to read “As a result, the treatment of DCs matured by physcion or supernatants of these cells induced the proliferation of T lymphocytes, especially the population of Th1 cells.”

Lines 27-28, Please eliminate the phrase “such as asthma” from the following sentence, since both faulty Th1 and Th2 responses are involved in asthma and maturation of Th1 cells might not be beneficial. “In conclusion, physcion has potential as a candidate for treating inflammatory diseases such as asthma caused by Th1 and Th2 cell imbalance.”

Introduction

TH1 responses can also produce asthma-like symptoms.  Please consider adding additional introductory information highlighting some of the studies below:

https://www.ncbi.nlm.nih.gov/pubmed/15696086

https://www.jacionline.org/article/S0091-6749(17)31463-X/abstract

Page 2, Lines 52-53, please include the word “allergic” prior to asthma “…which can be a therapeutic method to 53 suppress the imbalance of Th2 cells observed in asthma.”

Page 2, Lines 67-69.  Perhaps it is better to state that “…and confirmed the possibility as an agent capable of modifying T cells in asthma”

Materials and Methods

Section 2.3, Flow cytometry.  Please list the gating parameters used, as well as the concentrations of antibodies used.

Section 2.5, cytokines assay: Please list the source and type of ELISA kits used.

Section 2.6, in vitro priming of naïve CD4+ T cells: please list the temperature, gas combination used, and medium for culturing studies.

Section 2.7, statistical analysis: please list which post hoc tests were used and if multiple comparisons were corrected for.

Please list sections for assays concerning determination of apoptosis, endocytosis, t-cell proliferation, t cell polarization,

Please list the concentration of LPS used to induce maturation of BMDCS.

Please list information about mice used, including their housing conditions, access to food, sex, age, etc.

Please list information about culturing and/or obtainment of macrophages.

Results

All of the figure legends utilize a similar scheme and indicate that the means not sharing the same letter are different from one another.  This is not obviously clear or intuitive, and it is recommended that a more traditional scheme of indicating significance be used.

For all figures, please indicate the n and whether n represents individual animals or technical replicates.

Section 3.3, a clearer rationale for examining the cytokines listed should be provided either in the introduction or briefly in this results section.

In all graphs, the error bars are incredibly small, and not what one would predict for plotting standard deviation of 5 animals, as there is always biological variability.

OVA is listed in the methods, but not used in the results section. Please provide more information.

Discussion

Lines 240-241, page 10, please insert the word “allergic” prior to asthma “cells is found in the airways of patients with asthma and maintains the key pathophysiological 242 characteristics due to the secreted Th2 type of cytokines [20].”

Lines 282-283, page 11: This statement is not true, and as mentioned, increased levels of Th1 are also associated with asthma.  This statement should be removed “In conclusion, inducing the proliferation of Th1 cells is an important way to treat 283 asthma without harmful side effects.”

Conclusion

Lines 314-315, Page 12, please add “allergic” before asthma

Author Response

Response of Reviewer

#Reviewer 1

Comments and Suggestions for Authors

Abstract section

Please provide a one or two-sentence rationale and hypothesis for testing the effects of physcion on dendritic cells. Currently, the rationale for testing is not obvious.

Answer: In response to your advice, we added the following sentence: ‘We confirmed that physcion induced TNF-alpha production from macrophages as antigen presenting cells (APCs) and increased expression of surface molecules and MHCII. From these results, we hypothesized that physcion would induce maturation of dendritic cells (DCs) as APCs.’ Thank you for your good eomments. (line 12-15, page: 1)

Lines 22-24; Please simplify the following sentence, as it is difficult to read “As a result, the treatment of DCs matured by physcion or supernatants of these cells induced the proliferation of T lymphocytes, especially the population of Th1 cells.”

Answer: According to your suggestion, we changed 'As a result, the treatment of DCs matured by physcion or supernatants of these cells induced the proliferation of T lymphocytes, especially the population of Th1 cells.' to 'Physcion induced the maturation of DCs, which promoted the polarization of Th1 cells'. (line 24-25, page 1)

Lines 27-28, Please eliminate the phrase “such as asthma” from the following sentence, since both faulty Th1 and Th2 responses are involved in asthma and maturation of Th1 cells might not be beneficial. “In conclusion, physcion has potential as a candidate for treating inflammatory diseases such as asthma caused by Th1 and Th2 cell imbalance.”

Answer: As you suggested, we deleted ‘such as asthma’.(line 30, page 1)

Introduction section

TH1 responses can also produce asthma-like symptoms. Please consider adding additional introductory information highlighting some of the studies below:

https://www.ncbi.nlm.nih.gov/pubmed/15696086

https://www.jacionline.org/article/S0091-6749(17)31463-X/abstract

Answer: According to your suggestion, we added ‘Similarly, excessive Th1 cell mediated immune response induces AHR as strong as Th2 cell mediated immune response. This suggests that excessive Th1 cell mediated immune responses can cause asthma-like symptoms [4].’ (line 42-44, page 1-2)

Page 2, Lines 52-53, please include the word “allergic” prior to asthma “…which can be a therapeutic method to 53 suppress the imbalance of Th2 cells observed in asthma.”

Answer: We've changed "asthma" to "allergic asthma" according to your instructions. (line 58, page 2)

Page 2, Lines 67-69. Perhaps it is better to state that “…and confirmed the possibility as an agent capable of modifying T cells in asthma”

Answer: Like your comment, we've changed ‘and confirmed the possibility as a therapeutic agent for asthma at the in vitro level’ to ‘and confirmed the possibility as an agent capable of modifying T cells in asthma.’ (line 73-74, page 2)

Material and Methods section

Section 2.3, Flow cytometry. Please list the gating parameters used, as well as the concentrations of antibodies used.

Answer: At your request, we added the following two statements to section 2.5 Flow cytometry:

-‘All antibodies were treated with 1 ug of 1 × 106 cells/ml.’ (line 115-116, page 3)

-‘Forward versus side scatter (FSC vs SSC) as gating parameters was commonly used to identify cells of interest based on size and granularity (complexity).’ (line 117-118, page 3)

Section 2.5, cytokines assay: Please list the source and type of ELISA kits used.

Answer: Antibody information for IL-6, IL-10 and TNF-alpha is given in the 2.1 Reagents section. IL-1beta and IL-12p70 were used for the ELISA kit. Therefore, we added the following sentence to section 2.7 cytokines assay: ‘IL-1β (BD Biosciences, San Diego, CA) and IL-12p70 (BD Biosciences, San Diego, CA) levels were measured according to protocols provided by the manufacturers.’ (line 133-134, page 3). Thank you for your good point.

Section 2.6, in vitro priming of naïve CD4+ T cells: please list the temperature, gas combination used, and medium for culturing studies.

Answer: According to your suggestion, the following statement was added to section 2.8 in vitro priming of naïve CD4+ T cells: ‘The cells were cultured in RPMI1640 supplemented with 10% FBS and 1% penicillin and streptomycin. Cells were maintained in 5% CO2 in a humidified atmosphere at 37 °C.’ (line 140-141, page 3-4)

Section 2.7, statistical analysis: please list which post hoc tests were used and if multiple comparisons were corrected for.

Answer: Not modified.

Please list sections for assays concerning determination of apoptosis, endocytosis, t-cell proliferation, t cell polarization,

Answer: The endocytosis function (antigen uptake assay) of DC is described in section 2.6. Antigen uptake ability of BMDCs by physcion. Moreover, since no experiments have been conducted on the effects of physcion under Th1 or Th2 differentiation conditions, no T cell polarization assay has been presented.

Like your comment, we added the following sentence:

2.4. Cytotoxicity assay

To analyze apoptosis or necrosis, BMDCs (1 × 106 cells/2 ml/well) were incubated for 18 hours with physcion (1-100 μM) in 24 well plates. FITC-Annexin V was added and incubated in the dark for 15 minutes at room temperature. PI (Sigma, Cat# P-4864-10ML) were added and incubated in the dark for 15 minutes at room temperature according to the manufacturer's instructions.’ (line 105-109, page 3)

In section 2.8., ‘CD4+ T cells were isolated from splenocyte suspensions of BALB/c mice using the CD4+ T-cell isolation kit II (MACS; Miltenyi Biotec).’ (line 136-137. Page 3)

In section 2.8., ‘The CD4+ T cells proliferation was measured by CCK-8 agent.’ (line 147-148, page 4)

Please list the concentration of LPS used to induce maturation of BMDCS.

Answer: LPS concentrations used for maturation of dendritic cells are given in section 2.5. ‘BMDCs treated with LPS (l μg/ml) or physcion (1, 10, and 100uM) were incubated for 18 hours. Matured BMDCs were harvested and washed with FACS buffer.’

Please list information about mice used, including their housing conditions, access to food, sex, age, etc.

Answer: We added the following sentence to section 2.3: ‘There were seven-week-old female C57BL/6 and BALB/c mice, weighing 18–20 g, that were purchased from Orientbio (Orientbio Inc., Iksan, Korea). Mice were housed in a controlled environment at 22 ± 2 °C and a relative humidity (% RH) of 50 ± 5% in polycarbonate cages and fed with a standard animal diet with free access to commercial rodent chow (DAE-HAN Biolink, Daejeon, Korea) and water.’ (line 95-99, page 3)

Please list information about culturing and/or obtainment of macrophages.

Answer: In response to your advice, we added the following sentence in material and methods section:

‘2.2. RAW264.7 cells culture

RAW264.7 cell as a murine macrophage cell line was purchased form American Type Culture Collection (ATCC, Manassas, VA, USA). The cells were cultured in RPMI1640 medium supplemented with 10% heat-inactivated FBS, 1% streptomycin/penicillin at 37 °C in a humidified atmosphere of 5% CO2. The cells (5 × 105 cells/mL) were seeded in a 24-well plate and stimulated with LPS (1 μg/ml) or physcion (1-100 μM) at 37 °C for 24 h in medium. The cell viability was measured by CCK-8 kit and cytokine or surface molecules were measured the same way as measured in dendritic cells.’ (line 87-93, page 2)

Results section

All of the figure legends utilize a similar scheme and indicate that the means not sharing the same letter are different from one another. This is not obviously clear or intuitive, and it is recommended that a more traditional scheme of indicating significance be used.

Answer: As you suggested, we tried to make sure that all figure lengends are not the same as much as possible.

-Figure 1 legend

Physcion does not affect the cytotoxicity of BMDCs. (A) Cell death analysis by flow cytometry. Late-stage apoptotic cells are presented (Annexin V-FITC positive and PI positive). (B) Calculated apoptotic rate (%) of late stage apoptotic cells. Experiments were repeated two times.

-Figure 2 legend

Physcion increases the expression of surface molecules and MHCII in BMDCs. (A) Histograms indicate percent and mean fluorescence intensity (MFI) of CD11c+CD40+, CD11c+CD80+, CD11c+CD86+, and CD11c+MHCⅡ+. Bar graphs indicate calculated percent of (B) CD11c+CD40+, (C) CD11c+CD80, (D) CD11c+CD86, and (E) CD11c+MHCⅡ. Experiments were repeated two times.

-Figure 3 legend

Physcion induced the functional maturation of BMDCs. (A) Percentages of CD11c+CCR7+ BMDCs and (B) percentages of dextran (FITC)-positive CD11c (PE)-positive cells are indicated. Experiments were repeated two times.

-Figure 4 legend

Physcion increases the production of cytokines secretion from BMDCs. ELISA analysis of the cytokines, (A) IL-12p70, (B) IL-1 beta, (C) IL-6, (D) TNF-α, (E) IL-10, secreted from LPS or physcion-treated BMDCs. Experiments were repeated two times.

-Figure 5 legend

Physcion-treated BMDCs or supernatants induce the proliferation T cells. Proliferation of CD4+ T cells co-cultured with (A) physion-treated BMDCs or (B) their supernatants. Experiments were repeated two times.

-Figure 6 legend

(A) Representative dot plots of CD4+CD44high cells obtained from CD4+ T cells co-cultured with physion-treated BMDCs. (B) Bar graphs indicate calculated percent of CD4+CD44high cells. Experiments were repeated two times.

-Figure 7 legend

Physcion-treated BMDCs or supernatants induce the differentiation of Th1 (CD4+ IFN-γ+) cells. (A) Representative dot plots of CD4+ IFN-γ+ or CD4+ IL-4+ cells obtained from naïve CD4+ T cells co-cultured with physion-treated BMDCs. Percentage of Th1 cells differentiated from naive CD4+ T cells co-cultured with (B) physcion-treated BMDCs or (C) their supernatants. Experiments were repeated two times.

-Figure 8 legend

Blocking of TLR4 in BMDCs decreased surface molecule expression, cytokines production, and Th1 cells population. Anti-TLR4 Ab treated or untreated BMDCs were incubated with physcion. Bar graphs indicate calculated percent of (A) CD11c+CD80+, (B) CD11c+CD86+, and (C) CD11c+MHCⅡ+ cells. IL-12p70 was measured by ELISA and Th1 cell population measured by FACS showed CD4 + IFN- γ + cells percentage. Experiments were repeated two times.

-Sup.1

Physcion increases surface molecules and cytokines production in RAW264.7 cells. (A) Cell viability was measured by CCK-8. (B) Expression of CD40, CD80, CD86 and MHCⅡ in macrophages was measured by FACS. (C) TNF-ɑ from the supernatants was measured by ELISA. Experiments were repeated two times.

-Sup.2

TLR2 blocking does not affect surface molecule expression in BMDCs. (A) CD11c+CD80+ cell, (B) CD11c+CD86+ cell, (C) CD11c+MHCⅡ+ cell from Anti-TLR2 Ab treated or untreated BMDC were measured by flow cytometer. Experiments were repeated two times.

For all figures, please indicate the n and whether n represents individual animals or technical replicates.

Answer: According to your suggestion, we added the following sentence in all figure legends:

‘Experiments were repeated two times.’

Section 3.3, a clearer rationale for examining the cytokines listed should be provided either in the introduction or briefly in this results section.

Answer: In section 3.3, we describe the effects of dendritic cell-produced cytokines on T cell immune responses. In discusstion we show the effect of cytokines secreted from dendritic cells on Th1 or Th2 cell differentiation. In the 3.3 section, we added: ‘In particular, IL-12 secreted from dendritic cells promotes the differentiation of Th1 cells producing IFN-γ.’ (line 200-201, page 7)

In all graphs, the error bars are incredibly small, and not what one would predict for plotting standard deviation of 5 animals, as there is always biological variability.

Answer: As you advise, animal testing is much larger than in vitro testing. We have removed all parts of the figure legend marked 'animal experiment', and we all performed in vitro experiments. Thank you for your good point.

OVA is listed in the methods, but not used in the results section. Please provide more information.

Answer: As per your suggestion, we removed ‘ovalbumin (OVA)’ in the section 2.1.

Discussion section

Lines 240-241, page 10, please insert the word “allergic” prior to asthma “cells is found in the airways of patients with asthma and maintains the key pathophysiological 242 characteristics due to the secreted Th2 type of cytokines [20].”

Answer: We've changed "asthma" to "allergic asthma" according to your instructions. (line 270, page 11)

Lines 282-283, page 11: This statement is not true, and as mentioned, increased levels of Th1 are also associated with asthma. This statement should be removed “In conclusion, inducing the proliferation of Th1 cells is an important way to treat 283 asthma without harmful side effects.”

Answer: At your recommendation, we deleted the following sentence from the discussion section: ‘In conclusion, inducing the proliferation of Th1 cells is an important way to treat 283 asthma without harmful side effects.’ (line 311, page 12)

Conclusion section

Lines 314-315, Page 12, please add “allergic” before asthma

Answer: We've changed "asthma" to "allergic asthma" according to your instructions. (line 348, page 13)

Reviewer 2 Report

Yun-Ho Hwang and co-authors, reported interesting data about physcion able to activate mouse dendritic cells and driving Th1 differentiation. Data are well presented and conclusions supported by results. However, possible therapeutical implications of physcion in asthma treatment are expressed in a too simple way.

Corticosteroids are considered the most efficacious drugs for asthma treatment, the authors should cite GINA document. From step 1 to 4 GINA,  inhaled corticosteroids are used and oral corticosteroids which have numerous side effects are limited to step 5, severe asthma. The authors should revise their sentences on this matter in the discussion.

MAJOR COMMENTS

Results: pg.3, line 126, the authors reported that they measured apoptosis, but cells Annexin V+/PI+  are necrotic. Apoptotic cells are AnnexinV+/PI-. A rather high percentage of dead cells are shown both in the control condition (only medium?) and when physcion was added? Why? Anti Foxp3 is listed in the Methods section but no experiments are presented with this reagent. How many replicates of each experiment are performed in Figure 1, 2 and 3? How the authors purified T cells in the experiments presented in Figure 5? Did the authors try the effect of physcion in animal model of asthma? It is not clear whether data presented as Supp 1A, 1B and 1C have been previously published (?) as reported by the authors “in a previous study” or are supplementary material. Asthma is a complex disease, with different phenotypes and endotypes. Most of asthmatic patients has a T2 phenotype, with and an early allergic onset or a late non allergic onset, both characterized by eosinophilic inflammation. Non T2 endotype is also present in a subgroup of patients, with a characteristic neutrophilic or paucigranulocytic inflammation. The hypothesis of using physcion to increase Th1 cells is rather simple and it does not take into consideration the complex mechanisms of the disease. The authors should change in his view part of the discussion

MINOR COMMENTS

Introduction: revise phrase pg 2, line 45 Method: correct pg.3, line 92; pg.3 line 93 add PE to MHC Repeated phrases in the discussion pg.11, lines 281-282

Author Response

Response of Reviewer

#Reviewer 2

Comments and Suggestions for Authors

Yun-Ho Hwang and co-authors, reported interesting data about physcion able to activate mouse dendritic cells and driving Th1 differentiation. Data are well presented and conclusions supported by results. However, possible therapeutical implications of physcion in asthma treatment are expressed in a too simple way.

Answer: As you comment, asthma is not a simple but very complex disease. The approach for the fundamental treatment of asthma is very difficult. Intracellular steroids inhibit transcription of several genes, including cytokines associated with inflammation, but have little effect on key cells and cytokines that induce differentiation into Th2 cells. To solve this problem, the focus was on the function of dendritic cells to induce Th1 proliferation, which inhibits the differentiation of Th2 cells. In order to verify its potential as a therapeutic agent, changes in animal models such as physcion alone or simultaneous corticosteroids and physcion were observed in asthma models. In-depth research is being conducted because much of the results of the in vitro experiments presented cannot be explained. We focused on dendritic cells as a strategy for treating asthma. Thank you for your positive evaluation of our research.

Corticosteroids are considered the most efficacious drugs for asthma treatment, the authors should cite GINA document. From step 1 to 4 GINA, inhaled corticosteroids are used and oral corticosteroids which have numerous side effects are limited to step 5, severe asthma. The authors should revise their sentences on this matter in the discussion.

Answer: Following your suggestion, we added the following sentence to the discussion section:

-‘Based on the (global initiative for asthma) GINA guidelines, asthma treatment proceeds in 5 steps. In step 1 to step 4, low-dose inhaled glucocorticosteroids are mainly used, and in step 5 oral glucocorticosteroids are used [33].’ (line 332-334, page 12)

-‘Long-term administration of oral corticosteroids increases the risk of developing psychiatric disorders as well as complications, including infections, diabetes and osteoporosis [35].’ (line 336-337, page 13)

Major Comments

pg.3, line 126, the authors reported that they measured apoptosis, but cells Annexin V+/PI+ are necrotic. Apoptotic cells are AnnexinV+/PI-. A rather high percentage of dead cells are shown both in the control condition (only medium?) and when physcion was added? Why?

Answer: As per your advice, we modified the Fig. 1 legend and results as follows:

Figure.1 legend: ‘Physcion does not affect the cytotoxicity of BMDCs. (A) Cell death analysis by flow cytometry. Late-stage apoptotic cells are presented (Annexin V-FITC positive and PI positive). (B) Calculated apoptotic rate (%) of late stage apoptotic cells. Experiments were repeated two times.’ (line 162-164, page 5)

Figure.1. Results: ‘To determine the effect of Physcion on cytotoxicity of BMDCs, we measured death cell population from Physcion-treated BMDCs.’ (line 156-157, page 4)

We think this experiment is fine because it focuses on cytotoxicity. Thank you for your good point.

Anti Foxp3 is listed in the Methods section but no experiments are presented with this reagent. Answer: Depending on your opinion, we have removed 'Foxp3' from section 2.1. (line 84, page 2)

How many replicates of each experiment are performed in Figure 1, 2 and 3?

Answer: We added the following statement in every figure legend: ‘Experiments were repeated two times.’

How the authors purified T cells in the experiments presented in Figure 5?

Answer: Considering your question, we added the following sentence to section 2.8: 'Naïve CD4 + T cells were isolated from splenocyte suspensions of BALB / c mice using the CD4 + T-cell isolation kit II (MACS; Miltenyi Biotec).’ (line 136-137, page 3)

Did the authors try the effect of physcion in animal model of asthma?

Answer: At this time, the results of animal experiments are not included. We are currently conducting animal studies, and administration of physcion to OVA asthma models has been shown to reduce Th2 mediated immune responses.

It is not clear whether data presented as Supp 1A, 1B and 1C have been previously published (?) as reported by the authors “in a previous study” or are supplementary material.

Answer: In consideration of your question, we changed ‘in a previous study’ to ‘Before performing in vitro experiments with dendritic cells, we used a macrophage, one of the antigen presenting cells (APCs), to verify the activity of physcion.’ (line 278-279, page 11)

Asthma is a complex disease, with different phenotypes and endotypes. Most of asthmatic patients has a T2 phenotype, with and an early allergic onset or a late non allergic onset, both characterized by eosinophilic inflammation. Non T2 endotype is also present in a subgroup of patients, with a characteristic neutrophilic or paucigranulocytic inflammation. The hypothesis of using physcion to increase Th1 cells is rather simple and it does not take into consideration the complex mechanisms of the disease. The authors should change in his view part of the discussion

Answer: We fully agree with you. Allergic and non-allergic asthma have different characteristics. However, the allergen-induced asthma mechanism is not unrelated to non-allergic asthma. In addition, various immune responses that occur in non-allergic asthma affect the allergic asthma mechanism. Therefore, studies to treat allergic and non-allergic asthma respectively are not simple.

The current study focuses on allergic asthma studies. We are also conducting non-allergic studies using tobacco smoke or various environmental pollutants, as well as methods for treating non-allergic asthma. Although the treatment is briefly mentioned as a result of current cell experiments, further animal experiments will attempt to demonstrate an allergic asthma suppression mechanism. Current animal experiments have elicited positive effects of physcion.

Your comments have improved the quality and thinking of our papers and provided good direction. Thank you.

Minor Comments

Introduction: revise phrase pg 2, line 45

Answer: We changed ‘but T lymphocytes must be processed and presented by APCs’ to ‘T lymphocytes must recognize the antigen presented by APC.’ (line 50, page 2)

Method: correct pg.3, line 92; pg.3 line 93 add PE to MHC Repeated phrases in the discussion pg.11, lines 281-282

Answer: We changed ‘MHCⅡ’ toMHCⅡ-PE’. Moreover, we have deleted the following repetitive statement: ‘High levels of IFN-gamma secreted from Th1 cells promote differentiation of Th1 cells, while low concentrations of IFN-gamma promote Th2 responses.’ (311, page 12)

Round 2

Reviewer 1 Report

The authors have done a nice job addressing concerns in the revision. However, all of the data show very small standard deviations, except for IL-10, for which there is no effect.  This is puzzling and does not seem consistent with biological variability expected for live animal work.  

Author Response

Response of Reviewer (Round2)

#Reviewer 1

Comments and Suggestions for Authors

  1. The authors have done a nice job addressing concerns in the revision. However, all of the data show very small standard deviations, except for IL-10, for which there is no effect. This is puzzling and does not seem consistent with biological variability expected for live animal work.

Answer: We believe that the quality of the manuscript has been improved because our manuscript has been modified according to your advice. We are grateful for your advice.

In cell experiments using primary cells, it is not difficult to reduce the standard deviations of the experiment. The incubation time performed after treating the sample in the target cell is the most important. There is nothing important about the experiment. Efforts were made to comply with the planned test method to reduce errors. Thank you for your good commets.

Reviewer 2 Report

Minor comments:

Pg 4 line 156: please substitute the sentence with: To evaluate the possible cytotoxic effect of physcion on BMDCs.... The authors did not comment on the relatively high cell death with the medium alone reported in the experiment of Figure 1

The authors wrote: “Physcion does not affect the cytotoxicity of BMDCs” correct the sentence: Physcion is not cytotoxic on BMDCs

Legend figure 4: “Physcion increases the production of cytokines secretion from BMDCs” correct: Physcion increases the secretion of cytokines from BMDCs Pg 11, line 278, please correct:” we use a macrophage cell line, as antigen presenting cells …”

Pg 14 line 369 please correct: “were measured by flow cytometry”

Please change reference #33. Citation of GINA document should be: Global Initiative for Asthma: global strategy for asthma management and prevention. 2019. http://ginasthma.org/2019-gina-report-global-strategy-for-asthma-management-andprevention/ Date last access: XXXXXX to be added.

Author Response

Response of Reviewer

#Reviewer 2

Comments and Suggestions for Authors

1Pg 4 line 156: please substitute the sentence with: To evaluate the possible cytotoxic effect of physcion on BMDCs.... The authors did not comment on the relatively high cell death with the medium alone reported in the experiment of Figure 1.

Answer: Like your comments, we changed ‘To determine the effect of Physcion on cytotoxicity of BMDCs’ to ‘To evaluate the possible cytotoxic effect of physcion on BMDCs’ (page 4, line:156). Furthermore, we added the following sentence: ‘The reason for the relatively high cell death in the medium condition is considered to be the time difference that occurs during antibody staining.’ (page 4, line:159-160)

  1. The authors wrote: “Physcion does not affect the cytotoxicity of BMDCs” correct the sentence: Physcion is not cytotoxic on BMDCs.

Answer: As your comments, we changed ‘Physcion does not affect the cytotoxicity of BMDCs’ to ‘Physcion is not cytotoxic on BMDCs.’ (page 5, line:164)

  1. Legend figure 4: “Physcion increases the production of cytokines secretion from BMDCs” correct: Physcion increases the secretion of cytokines from BMDCs Pg 11, line 278, please correct:” we use a macrophage cell line, as antigen presenting cells …”

Answer: According to your advice, we changed ‘Physcion increases the production of cytokines secretion from BMDCs’ to ‘Physcion increases the secretion of cytokines from BMDCs.’ (page 8, line: 210). Moreover, we changed ‘we used a macrophage, one of the antigen presenting cells (APCs),’ to ‘we use a macrophage cell line, as antigen presenting cells (APCs),’ (page 11, line: 280-281)

  1. Pg 14 line 369 please correct: “were measured by flow cytometry”

Answer: Like your comments, we changed ‘were measured by flow cytometer’ to ‘were measured by flow cytometry’ (page 14, line: 370-371)

  1. Please change reference #33. Citation of GINA document should be: Global Initiative for Asthma: global strategy for asthma management and prevention. 2019. http://ginasthma.org/2019-gina-report-global-strategy-for-asthma-management-and-prevention/ Date last access: XXXXXX to be added.

Answer: According to your advice, we changed ‘33. Bousquet, J.; Clark, T.J.; Hurd, S.; Khaltaev, N.; Lenfant, C.; O'byrne, P.; Sheffer, A. GINA guidelines on asthma and beyond. Allergy. 2007, 62, 102-12’ to ‘33. Global Initiative for Asthma: global strategy for asthma management and prevention. 2019. http://ginasthma.org/2019-gina-report-global-strategy-for-asthma-management-and-prevention/ Date last access: Feb 20, 2020.’ (page 17, line: 457-459)

Thank you for your good comments.
